# Galectins and Liver Diseases

**DOI:** 10.3390/ijms26020790

**Published:** 2025-01-18

**Authors:** Shima Mimura, Asahiro Morishita, Kyoko Oura, Kei Takuma, Mai Nakahara, Tomoko Tadokoro, Koji Fujita, Joji Tani, Hideki Kobara

**Affiliations:** Departments of Gastroenterology and Neurology, Faculty of Medicine, Kagawa University, Kita-gun, Takamatsu 761-0793, Kagawa Prefecture, Japan

**Keywords:** galectin, immune response, apoptosis, angiogenesis, fibrosis, biomarker, gut–liver axis, hepatocellular carcinoma

## Abstract

Galectins are widely distributed throughout the animal kingdom, from marine sponges to mammals. Galectins are a family of soluble lectins that specifically recognize β-galactoside-containing glycans and are categorized into three subgroups based on the number and function of their carbohydrate recognition domains (CRDs). The interaction of galectins with specific ligands mediates a wide range of biological activities, depending on the cell type, tissue context, expression levels of individual galectin, and receptor involvement. Galectins affect various immune cell processes through both intracellular and extracellular mechanisms and play roles in processes, such as apoptosis, angiogenesis, and fibrosis. Their importance has increased in recent years because they are recognized as biomarkers, therapeutic agents, and drug targets, with many other applications in conditions such as cardiovascular diseases and cancer. However, little is known about the involvement of galectins in liver diseases. Here, we review the functions of various galectins and evaluate their roles in liver diseases.

## 1. Introduction

The discovery of galectin dates back to 1975, when Teichberg et al. discovered a low-molecular-weight (14–16 kDa) erythrocyte aggregate that could be inhibited by β-galactoside in various animal tissues, including those of electric eels. As far as we are aware, this is the first documented report on galactoside-binding lectins (galectins) [1]. Lectins are defined as “glycan-binding proteins that are neither enzymes nor antibodies” and are present in various tissues, where they play a key role in influencing cell fate [2]. Among endogenous soluble lectins, galectins are believed to engage with cell-surface sugar chains in the extracellular environment, modulate signal transduction pathways, and influence cell fate decisions [3,4]. Galectins were originally identified in vertebrates; however, they are widely distributed in the animal kingdom, from sponges to mammals, with galectin-like domains also reported in viruses and plants [5].

Unlike many intracellular molecules and pathways that specialize in specific functions, galectins perform diverse immunological roles [6]. Many galectins can move between intracellular compartments (e.g., the nucleus, cytoplasm, and organelles) and are released into the extracellular environment [7]. Afterward, they acquire different roles in response to diverse microenvironmental stimuli, such as hypoxia, nutritional conditions, intra- and extracellular pH levels, cytokine environments, and inflammatory or immunosuppressive signals. Galectins have attracted increasing attention as diagnostic and therapeutic targets for diseases, such as fibrosis, cancer, and inflammatory conditions [8,9].

However, little is known about the involvement of galectins in liver disease. Although previous works providing an overview of galectins and their role in the liver are available, their content remains limited. In this review, we aim to provide the most comprehensive explanation of liver diseases to date, incorporating emerging topics such as galectins in liver diseases and the gut–liver axis, as well as the latest insights into the roles of galectins.

## 2. Galectin Family

### 2.1. Structure of Galectins

Galectins are a group of soluble lectins that specifically bind to glycans containing β-galactosides. Galectins found in the human genome were classified by the HUGO Gene Nomenclature Committee into galectin-1, -2, -3, -4, -7, -7B, -8, -9, -9B, -9C, -10, -12, -13, -14, and -16 (gene group: Galectins (LGALS)) [10]. All galectins contain a conserved carbohydrate recognition-binding domain (CRD) of approximately 130 amino acids [6,11,12,13]. However, individual galectins display distinct specificities in their carbohydrate-binding properties [14,15,16,17]. Galectins are classified into three subgroups, proto-, tandem-repeat, and chimeric type, depending on the number and function of their CRDs (Figure 1a–c) [18].

Prototypic galectins possess a single CRD and have the ability to dimerize through noncovalent interactions (e.g., Gal-1, -2, -5, -7, -10, -11, -13, -14, -16). These galectins assemble into homodimers, with two identical CRDs joined by noncovalent electrostatic interactions that are concentration-dependent and occur independently of ligand binding [15,19].

Tandem-repeat type galectins have two distinct CRDs, each with different sugar-binding capacities, connected by 5–50 amino acids (Gal-4, -6, -8, -9, and -12) [19].

The only member of the chimeric galectins is Gal-3, which has a unique structure distinct from other galectin family members and can form pentamers [15,16,20,21]. Chimeric galectins are distinguished by having a single C-terminal CRD and an extended non-lectin N-terminal domain (approximately 120 amino acids) abundant in proline, glycine, and tyrosine residues, potentially facilitating Gal-3 oligomerization [19]. Gal-3 has a large number of structurally and functionally diverse biological ligands [22]. Gal-3 is the only molecule capable of forming pentamers. It exhibits unique multifunctional biological properties and plays a crucial role in many physiological and pathological processes [23].

### 2.2. Functions of Galectin Structures

Galectins are a crucial family of β-galactoside-binding lectins involved in regulating interactions between cells as well as between cells and the extracellular matrix [20,24,25]. Their interaction with specific ligands triggers diverse biological activities, depending on the cell type, tissue context, individual galectin expression levels, and receptors involved. For instance, Gal-7 is mainly localized in the skin; Gal-10 is strongly expressed in eosinophils. Additionally, some galectins exhibit more restricted tissue localization, such as Gal-12 in adipose tissue [26]. Gal-4 is typically expressed in gastrointestinal epithelial cells [27]. Like many cytokines, galectins exhibit diverse actions that overlap with immunomodulatory activities. For cytokines, this diversity and overlap can be partly attributed to the fact that they share common signaling molecules. Regarding galectins, this diversity and redundancy may arise from their ability to bind ligands multivalently, as soluble galectins can cross-link ligands on the cell surface via at least three different (but not mutually exclusive) pathways [28]. Because of their varied structures, galectins are capable of binding to a broad array of ligands, influencing multiple cellular processes such as adhesion, aggregation, angiogenesis, apoptosis, autophagy, proliferation, and metastasis [15,16,29].

## 3. Functions of Galectins

### 3.1. Regulation of Immune Response

Galectins influence various immune cell processes via both intracellular and extracellular mechanisms [30]. They play significant roles in both innate and adaptive immune responses [31], regulating immune cell homeostasis during adaptive immune responses [32,33,34].

Galectins have diverse effects on the cells involved in innate immune responses [35,36]. Galectins play several important roles during the inflammatory response. They influence the ability of innate immune cells to respond to chemotactic gradients and migrate across the endothelium. Additionally, galectins are involved in the synthesis and release of pro-inflammatory or anti-inflammatory cytokines, as well as in recognizing and eliminating pathogens and damaged cells [37].

Gal-1 induces the proliferation of activated T cells and B cells [38], macrophages [39], regulatory T cells (T-regs cells), and dendritic cells (DCs) [40]. Gal-1 also functions as a negative regulatory checkpoint for receptors involved in signal transduction [41,42,43]. Specifically, Gal-1 promotes apoptosis in activated T cells by binding to the CD45 receptor. Notably, activated T cells can produce Gal-1 through a signaling pathway dependent on MEK1/ERK and p38 MAP kinases, indicating a potential autocrine suicide mechanism to terminate effector immune responses [44]. Additionally, Gal-1 is involved in regulating Th1- and Th17-mediated responses, steering the immune response towards a Th2-type profile [30,45].

Gal-3 helps regulate key biological processes, such as regeneration, cell migration, and immune responses [46,47]. Recent research has shown that Gal-3 plays a significant role in regulating immune responses by inhibiting T cell activity [46,48,49,50,51]. Specifically, extracellular Gal-3 influences the formation of immune synapses, reducing T cell activation. Additionally, intracellular Gal-3 interacts with different membrane lipids and proteins, impacting processes like endocytosis and signaling through T cell receptors [52,53].

Gal-9 is abundantly expressed in several immune system tissues, including bone marrow, spleen, thymus, and lymph nodes. Gal-9, when released by activated T cells, induces eosinophil chemotaxis, activation, oxidative responses, degranulation and maturation of monocyte-derived DCs [41,54,55]. DCs maturation is evidenced by increased expression of Th1 cytokines and co-stimulatory molecules such as HLA-DR, CD83, CD80, CD54, and CD40. Thus, Gal-9 contributes to innate immunity by maturing dendritic cells [56]. These mature DCs subsequently migrate to the lymph nodes, where they trigger T cell activation. In addition, Gal-9 acts as a chemotaxis factor for eosinophils, regulating the signal-dependent chemotaxis of neutrophils and promoting phagocytosis [57,58]. In monocytes, intracellular Gal-9 triggers the transcription of inflammatory cytokines, including IL-1α, IL-1β, and IFN-γ [59]. Conversely, Gal-9 supports the expansion of immunosuppressive macrophages [60,61].

### 3.2. Apoptosis

Apoptosis refers to a form of programmed cell death, distinguished by specific morphological changes triggered by both internal and external stresses [62]. Apoptosis occurs during normal tissue development and morphogenesis [63]. Once apoptosis is initiated, the cell membrane remains stable, preventing the release of cytokines and other pro-inflammatory substances from the disrupted cells, thereby weakening inflammation and tissue damage [64].

In vivo studies have shown that extracellular Gal-1 binds to glycan molecules expressed on the surface of eosinophils, induces cell death via apoptosis, and inhibits cell migration [65]. Gal-1 also induces T cell apoptosis [66]. Gal-9, a tandem-repeat member of the galectin family, primarily functions as an anti-inflammatory lectin and promotes Th1 and Th17 cell apoptosis by interacting with T cell immunoglobulin domain and mucin domain protein 3 (Tim-3) [67]. Furthermore, Gal-9 contributes to the disruption of apoptotic processes during oncogenesis and directly triggers apoptosis in gastrointestinal cancer cells [9].

### 3.3. Angiogenesis

Galectins are multifaceted molecules that exhibit immunomodulatory and pro-angiogenic functions [67]. Gal-1 binds to VEGFR2 and neuropilin-1 in endothelial cells, promoting angiogenesis [68] and mimicking the effects of VEGF. The inhibition of Gal-1 binding to complex N-glycans attenuates both cancer-induced immunosuppression and angiogenesis. Furthermore, the early targeting of this lectin during tumor progression promotes vascular normalization, alleviating tumor hypoxia and increasing the influx of immune cells into the tumor microenvironment [69]. On the other hand, Gal-3 promotes angiogenesis by binding to integrins αvβ3 [70]. In addition, Gal-8 enhances angiogenesis by cross-linking activated leukocyte cell adhesion molecules and supports lymphangiogenesis by interacting with podoplanin and integrins [71]. These interactions between galectins and glycans affect tumor angiogenesis by engaging different receptors and signaling pathways.

### 3.4. Fibrosis

Fibrosis represents the final common pathway in chronic tissue damage. Persistent inflammation, fibrosis, disruption of tissue architecture, and organ dysfunction are key features of the pathology of numerous human diseases and leading contributors to global morbidity and mortality [72]. Fibroblasts and myofibroblasts are key cells involved in initiating and perpetuating organ scarring [73]. Gal-3 plays a role in triggering and amplifying acute inflammatory responses by recruiting macrophages to injury sites and sustaining chronic inflammation via pro-inflammatory signaling pathways. This can lead to fibrosis due to unresolved inflammation and abnormal tissue repair. In an experimental model of liver fibrosis, Gal-3 expression was closely linked both spatially and temporally with myofibroblast activation and collagen deposition. Gal-3 functions as an immediate early gene, rapidly upregulated in response to tissue injury [74]. Gal-3 is critically involved in the pathological remodeling of the myocardium after cardiac injury, chronic stress, or inflammation, ultimately contributing to myocardial fibrosis—a key feature of atrial fibrillation and heart failure [75,76,77,78,79].

## 4. Galectins and Medicine

### 4.1. Galectins as Disease Biomarkers

In 2014, the U.S. and FDA included Gal-3 on their list of validated cardiovascular biomarkers [78]. Increased serum levels of Gal-3 have been detected in nearly all forms of cardiovascular diseases, and its prognostic significance for various clinical outcomes has been thoroughly investigated [80,81,82]. An increase in Gal-3 levels has been shown to be positively correlated with the occurrence of heart failure [83,84]. Gal-3 serves as a valuable prognostic marker for both acute and chronic heart failure [85]. Higher Gal-3 levels indicate an increased risk of all-cause and cardiovascular mortality, as well as a higher risk of complications [86].

Galectin expression has been observed in various types of cancer. Gal-9 is expressed in a wide range of tumors, including those of the liver, small intestine, thymus, kidney, spleen, lung, heart, and skeletal muscle [87]. Gal-3 is expressed in the thyroid gland [88,89,90], stomach [91], colorectal tissue [92], and pancreatic cancers [93]. It is also a diagnostic and prognostic marker for breast cancer. Gal-3 expression is redistributed from the luminal epithelial cells in the more invasive ducts to the peripheral epithelial cells [90]. In melanoma, Gal-3 accumulates in the nuclei of melanocytes as the disease progresses [94]. Recently, Gal-4 was found to be involved in the peritoneal metastasis of malignant gastric cancer cells by interacting with various molecules, such as c-MET and CD44, and modulating glycosylation processes [95,96,97].

Galectins have great potential as biomarkers. Gal-3 is easily secreted onto the cell surface and into body fluids such as serum and urine, and is also released from injured and inflammatory cells. Therefore, Gal-3 may be used as a sensitive diagnostic or prognostic biomarker under various pathological conditions [18]. However, galectins are considered limited biomarkers due to large gaps in sensitivity and specificity between different subtypes, and their value is further limited by the lack of population data and variability in genetic and environmental factors [98].

### 4.2. Application as a Therapeutic Target

Galectins play important roles in tumor development, progression, and metastasis [93,99]. Galectins are widely regarded as potential therapeutic targets, leading to the development of numerous galectin inhibitors, some of which have been successfully evaluated in clinical trials [100]. Endothelial cells are particularly appealing targets as they are directly accessible through the bloodstream, allowing circulating drugs to reach them easily [101]. Furthermore, targeting galectins in the vasculature may provide dual benefits by inhibiting tumor angiogenesis while simultaneously alleviating immunosuppression. This dual effect is supported by numerous studies demonstrating that targeting galectins inhibits tumor progression by reducing tumor angiogenesis and enhancing anti-tumor immune responses [102,103,104,105].

Recently, therapies targeting immune checkpoints have gained momentum, unleashing anti-tumor immunity and providing significant clinical benefits to patients with various malignancies. Galectins are increasingly recognized as critical regulatory checkpoints in immune evasion, contributing to T cell exhaustion, reducing T cell survival, enhancing the proliferation of regulatory T cells, suppressing natural killer cell activity, and driving bone marrow cells toward immunosuppressive phenotypes [13,67,106].

## 5. Expression Patterns and Roles of Galectins in the Liver

Gal-1, -3, and -9 are particularly well-reported galectins expressed in the liver. Gal-1 is widely expressed in the liver, especially in activated hepatic stellate cells, where it is expressed at high levels [107]. In a normal liver, Gal-1 expression is mainly observed in endothelial cells, lymphocytes, and hepatocytes around the central vein [108]. It has been reported that Gal-1 expression is induced during liver regeneration [108].

Gal-3 expression is low in normal hepatocytes but increases in cirrhosis and hepatocellular carcinoma [94]. It has been shown that Gal-3 expression is increased during liver inflammation and fibrosis [109]. Gal-3 plays a role in promoting hepatic stellate cell activation and proliferation [110].

Gal-9 is highly expressed in the liver, especially in Kupffer cells [111,112]. It plays an important role in maintaining immune homeostasis in the liver [111]. In liver tissue from patients with acute liver failure, Gal-9 has been reported to be localized in regenerating areas and colocalized with Kupffer cells. Kupffer cells play an important role in hepatic immunoregulation [113]. Kupffer cells function as a bridge between innate and adaptive immunity through the production of Gal-9 [113].

## 6. Liver Fibrosis

Liver fibrosis represents the final pathway of chronic liver injury, associated with the accumulation of extracellular matrix proteins [23]. Understanding liver fibrosis is important because the progression of liver fibrosis gradually reduces liver function and may ultimately lead to liver failure. Gal-3 plays a central role in the progression of liver fibrosis. The connection between Gal-3 and hepatic fibrosis was demonstrated when Gal-3-deficient mice exhibited resistance to toxin-induced liver fibrosis [23,72]. Gal-3 is secreted by activated hepatic stellate cells (HSCs) and inflammatory macrophages, and promotes fibroblast activation [114]. Gal-3 also plays a role in promoting the activation and proliferation of HSCs [23,114]. Gal-3 amplifies extracellular matrix production and fibrogenesis through autocrine signaling pathways [115]. Gal-3 promotes the phagocytosis of apoptotic cells by HSCs through interaction with integrin αvβ3 [114]. The phagocytosis of apoptotic cells induces the transformation of HSCs into myofibroblasts, collagen I production, and TGF-β production [114,116]. The activation of myofibroblasts and the production of procollagen by TGF-β rely on intracellular Gal-3. The overproduction of extracellular matrix molecules and collagen by activated HSCs leads to altered tissue architecture, further damage, and ultimately organ failure through the formation of fibrous ridges and bridging fibrosis [116] (Figure 2).

## 7. Galectins and the Gut–Liver Axis

Over the past decade, a growing number of studies have supported the association between gut microbiota dysbiosis and metabolic diseases such as metabolic dysfunction-associated steatotic liver disease [117]. The disruption of the complex crosstalk between the gut and the liver is often referred to as the “gut–liver axis” [117]. Galectins play important functions in both the gut and liver, and it has been suggested that they play a central role in interactions mediated by the gut–liver axis. In particular, Gal-1, -2, -3, -4, and -9 are expressed in the intestinal tract, each of which has different functions [118,119]. Gal-1 is mainly present in the lamina propria mucosal and suppresses excessive immune responses by promoting the induction of regulatory T cells and the apoptosis of inflammatory T cells [118]. Gal-3 is highly expressed in intestinal epithelial cells and is involved in interactions with intestinal bacteria and in maintaining intestinal barrier function [119]. When the intestinal barrier function is disrupted and substances derived from intestinal bacteria enter the liver via the portal vein, inflammation and fibrosis of the liver are promoted. Galectins are involved in this process [120]. Alterations in the gut microbiota may affect the gut barrier function and the inflammatory state of the liver via galectins. Abnormal expression of galectins has been reported in pathologies such as nonalcoholic fatty liver disease [121].

## 8. Galectins and Liver Diseases

### 8.1. Chronic Hepatitis B (CHB)

Monocytes that express both Gal-9+ and PD-L1+ were significantly elevated in patients with HBeAg-positive and HBeAg-negative CHB compared to healthy controls [122]. In CHB, the combined effect of high HBsAg levels with increased TNF-α, IL-4, and IL-1β induces pleiotropic effects including the upregulation of both Gal-9 and PD-L1 on monocytes. This leads to Gal-9-dependent immunological alterations and severely weakens the host immune response through PD-L1-induced suppression of antiviral cytokine release by HBV-specific T/B cells and NK cells [122]. The expression of Gal-9 and PD-L1 on monocytes varies throughout different stages of CHB and plays distinct roles in inhibiting immune responses. Targeting these molecules therapeutically could potentially enhance the immune defense against hepatitis B virus (Table 1).

In HBV-related HCC, positive Gal-9 expression was associated with lymph node metastasis, a high Ki-67 proliferation index, and poor prognosis. Both univariate and multivariate analyses showed that Gal-9 expression serves as an independent prognostic marker for HBV-related HCC [123,124]. The interaction between Tim-3 on Th1 and its ligand Gal-9 negatively regulates Th1-mediated immune responses. We showed that the Tim-3/Gal-9 signaling pathway mediates T cell senescence in HBV-associated HCC [125].

Gal-3, through its interaction with macrophages, can stimulate cytokine and chemokine production via CD98 [126]. It plays an important role in the maintenance of HBV replication and may contribute to the pathological processes that lead to chronic HBV infection. Furthermore, Gal-3 can stimulate fibrogenesis by decreasing IL-10 production [127,128,129].

### 8.2. Hepatitis C Virus (HCV) Infection

Gal-9 and other members of the galectin family are widely implicated in viral infections, though their exact roles remain unclear [130]. Gal-9 expression has been observed in hepatocytes and Kupffer cells from liver biopsies of patients with hepatitis C virus (HCV) infections [152]. The levels of circulating Gal-9 and its expression in the serum of patients with HCV showed a positive correlation with the continued presence of HCV infection and the progression of chronic liver disease [131]. The secretion of Gal-9 is promoted by IFN-α, which subsequently inhibits HCV infection [132].

HCV-positive patients exhibit significantly higher Gal-3 levels than HCV-negative patients [133]. However, Gal-3 levels in the serum of patients with HCV show no correlation with viral load, viral genotype, CRP, white blood cell count, or end-stage liver disease scores [153].

### 8.3. Metabolic Dysfunction-Associated Steatotic Liver Disease/Steatohepatitis (MASLD/MASH)

NOD-like receptor protein 3 (NLRP3) is a key molecule in inflammation research and its activation influences the progression of various inflammatory diseases. Gal-3 can activate the NLRP3 inflammasome via toll-like receptor 4 (TLR4), promoting inflammatory responses in hepatocytes. The CRD of Gal-3 regulates the TLR4/NLRP3 pro-inflammatory pathway, playing a role in the initiation of lipid imbalance and inflammation in the liver [134].

MASLD and its more severe subtype, MASH, are highly prevalent and strongly associated with obesity and type 2 diabetes [154]. Gal-3 levels are higher in obese and diabetic individuals and have been shown to increase in association with uninhibited glucose homeostasis. Additionally, Gal-3 is thought to regulate adipogenesis by promoting the differentiation of progenitor cells into mature adipocytes [135].

Advanced glycation end products (AGEs) are a heterogeneous group of glycosylated proteins that accumulate with aging [155]. Elevated AGE levels are observed in individuals with diabetes and liver cirrhosis [156]. In diabetic patients, Gal-3 exhibits both AGE-binding-dependent and -independent functions. The AGE-dependent roles of Gal-3 differ across organs, reflecting tissue-specific variations in the AGE receptor system, with the liver playing a central role in the clearance and detoxification of AGEs from the bloodstream. Consequently, most studies regard Gal-3 as a marker of inflammation and fibrosis. However, research using animal models of metabolic disorders suggests that increased Gal-3 expression may serve as an adaptive response to tissue damage, aiding in the resolution of inflammation and preventing the progression to chronic inflammation [136].

In addition to NK cell markers, there exists a subset of cells with a unique immunological profile, defined by the expression of invariant TCRs (Vα14-Jα18 in mice and Vα24-Jα18 in humans). These cells are known as NKT cells [157]. In the liver, Tim-3+ NKT cells are activated, and Gal-9 directly triggers their apoptosis, leading to NKT cell depletion during diet-induced steatosis. Additionally, Gal-9 interacts with Tim-3-expressing Kupffer cells to stimulate IL-15 secretion and promote NKT cell proliferation. Exogenous Gal-9 administration significantly improves diet-induced steatosis by modulating NKT cell activity in the liver. In conclusion, the Tim-3/Gal-9 signaling pathway plays a crucial role in regulating hepatic NKT cell homeostasis through activation-induced apoptosis and subsequent proliferation, thereby contributing to the development of MASH [137].

### 8.4. Alcohol Associated Liver Disease (ALD)

Patients with ALD, particularly those with a progressive form of the disease, exhibit significantly elevated supraphysiological plasma levels of soluble Tim-3 and its soluble ligand Gal-9 [138]. This increase is linked to higher levels of soluble Tim-3 ligands and membrane-bound Tim-3 expression in immune cells. While soluble Tim-3 can disrupt Tim-3-ligand interactions and enhance antimicrobial immunity, the elevated levels of soluble Tim-3-binding ligands in patients with ALD counteract this immunostimulatory effect [138].

### 8.5. Autoimmune Hepatitis (AIH)

AIH, histologically referred to as interfacial hepatitis, is a progressive inflammatory liver disease characterized by hypergammaglobulinemia, circulating autoantibodies, and flamboyant mononuclear cell infiltration [158]. CD4 effector lymphocytes play a central role in liver damage in AIH, with their proliferation and secretion of pro-inflammatory cytokine secretion (e.g., interferon γ [IFNγ]) being linked to the activity and severity of the disease [159]. In AIH, the degree of autoreactive CD4 T cell effector immune response correlates with the quantity and functional impairment of CD4 cells [150]. Gal-9 is a β-galactosidase-binding protein expressed by T-regs, and it plays a key role in their function. It inhibits Th1 immune response by binding to Tim-3 on CD4 effector cells. Decreased levels of Tim-3 in CD4(pos) CD25(neg) effector cells and Gal-9 in T-regs disrupt immunoregulation in AIH by making effector cells less susceptible to T-reg control and impairing the suppressive function of T-regs [139].

### 8.6. Primary Biliary Cholangitis (PBC)

PBC is a chronic autoimmune liver disease characterized by destructive lymphocytic inflammation targeting the small bile ducts, elevated serum levels of antimitochondrial antibodies (AMAs) specific to mitochondrial autoantigens, and a significantly higher prevalence in females [160]. Gal-3 contributes to the inflammatory process in PBC by directly interacting with NLRP3 and promoting inflammasome activation in hepatic macrophages. This activation leads to the production of pro-inflammatory cytokines, which compromise the integrity of biliary epithelial cells and cause tissue damage [140]. As a result, Gal-3 expression in biliary epithelial cells is upregulated during PBC, highlighting the active role of these cells in disease pathogenesis.

### 8.7. Hepatocellular Carcinoma and Galectins

Gal-3 is absent in normal hepatocytes, whereas it is highly expressed in cirrhotic liver cancer and hepatocellular carcinoma (HCC), indicating its potential involvement in the development of cirrhosis and HCC [141]. Analysis of the relationship between Gal-3 expression and microvessel density revealed that Gal-3 promotes angiogenesis in tumor cells. Additionally, Gal-3 regulates apoptosis in HCC cells through its influence on the caspase-3 signaling pathway [142]. Furthermore, Gal-3 suppresses tumor-reactive T cells, thereby enhancing tumor growth in mouse models treated with tumor-reactive T cells [143].

The urokinase plasminogen activator receptor (uPAR) is overexpressed in various human cancers and is often linked to poor clinical outcomes [161]. In HCC cells, silencing Gal-3 has been shown significantly decrease the mRNA and protein levels of uPAR, along with its downstream signaling pathway, by suppressing the MEK/ERK signaling pathway. This suppression may subsequently inhibit the proliferation, migration, and invasion of HCC cells [144] (Figure 3).

Gal-9 suppresses the proliferation of HCC cell lines by inducing cell apoptosis [145]. Moreover, Gal-9 enhances anti-tumor immunity by increasing the number of Tim-3-expressing DCs and CD8+ T cells through its interaction with Tim-3 [125]. While typical Gal-9 receptors, such as T-cell immunoglobulin and Tim-3, are absent on the surface of HCC cell lines. Gal-9-induced apoptosis is inhibited by lactose administration, suggesting that this receptor should be glycosylated by β-galactoside [145]. Additionally, interferon stimulates Gal-9 expression in HCC cells. This upregulation is associated with Gal-9 being a target of microRNA 22 (miR-22), with the antitumor effects of Gal-9 enhanced when its expression is restored in cells overexpressing miR-22 [146].

Recent studies have revealed that patients with high serum Gal-9 levels have a significantly shorter time to tumor recurrence [147]. Furthermore, serum Gal-9 has been identified as an independent predictor of HCC recurrence, even in patients with low AFP levels or at an early stage of the disease. High expression of Gal-9 has been shown to be involved in immune evasion and inflammatory signaling pathways. This was correlated with increased infiltration of exhausted CD8+ T cells, regulatory T cells, TAMs, and MDSCs. Interestingly, Gal-9 was predominantly expressed in macrophages rather than malignant cells. Similarly, high serum Gal-9 levels also reflected an immune-evasive microenvironment characterized by extensive CD163+ and FOXP3+ cell infiltration [147].

Gal-1 expression is elevated in HCC and increases further as the tumor progresses to advanced stages [162]. Gal-1 has been shown to interact with mRNA that preferentially binds to obstructive codons, thereby regulating angiogenesis [148]. Epithelial-mesenchymal transition (EMT) is widely recognized as a critical process in facilitating the dissemination of malignant hepatocytes during the progression of HCC; Gal-1 overexpression promotes HCC progression by inducing HCC cell EMT via PI3K/AKT cascade activation [149,150,151].

## 9. Application to Liver Disease Diagnosis

### 9.1. Liver Fibrosis Screening

It has been reported that serum Gal-9 levels increase with the progression of liver fibrosis. An increase of 10 pg/mL in serum Gal-9 concentration has been linked to a 3.90-fold higher likelihood of advancing to liver fibrosis [131]. Therefore, serum Gal-9 levels are expected to be a potential biomarker for liver fibrosis in patients with chronic liver disease.

Elevated Gal-3 levels in liver biopsies can help differentiate between F3/F4 and F0/F1 stages of fibrosis [163]. Furthermore, Gal-3 is linked to fibrotic zones within the liver [164]. Gal-3 is expressed in Kupffer cells during the progression of liver fibrosis, and more recently, Gal-3-related binding proteins have been reported as serum surrogate markers for evaluating liver fibrosis [18]. Thus, Gal-3 has the potential to serve as a useful biomarker for fibrogenic liver diseases leading to cirrhosis in both serum and biopsy samples [165]. Early evaluation and identification of liver diseases are clinically significant.

### 9.2. Early Detection and Prognosis of HCC

Serum Gal-3 levels were significantly higher in patients with HCC compared with healthy controls (mean difference = 3.06, 95% CI = 1.79–4.32, *p* < 0.001) [166]. High expression of Gal-1 and -3 in tissues of HCC patients has been shown to be associated with poor overall survival (Gal-1: HR = 1.94, 95% CI = 1.61–2.34, *p* < 0.001; Gal-3: HR = 3.29, 95% CI = 1.62–6.68, *p* < 0.001). On the other hand, high expression of Gal-4 and -9 has been reported to be associated with favorable overall survival in HCC patients (Gal-4: HR = 0.53, 95% CI = 0.36–0.79, *p* = 0.002; Gal-9: HR = 0.56, 95% CI = 0.44–0.71, *p* < 0.001) [166].

### 9.3. Early Diagnosis of Acute Liver Damage

Detecting a sudden increase in serum Gal-3 levels may be useful for early diagnosis and severity evaluation of acute liver damage [167]. Gal-3 is expressed on NKT cells in the liver, where it participates in interactions with DCs and plays a key pro-inflammatory role in acute liver injury [168].

## 10. Conclusions

Galectins are a family of proteins with significant potential in the medical field. Therapies targeting galectins are emerging as promising approaches for treating many diseases, including cancer, fibrosis, and cardiovascular conditions. Galectins are gaining prominence as diagnostic biomarkers, potentially contributing to early diagnosis and the advancement of personalized medicine. In the field of the liver, it may be possible to use time-dependent changes in serum galectin levels to assess the effectiveness of antiviral therapy and liver fibrosis treatment. In addition, serum galectin levels may be used to stratify the risk of disease progression in patients with chronic liver disease, allowing for more appropriate management and follow-up plans. These applications are expected to enable early detection and early intervention of liver diseases, leading to improved prognosis for patients. However, before diagnostic tools using galectins can be put to practical use, validation through large-scale prospective studies, standardization of measurement methods, optimization of cutoff values, etc., are required. While galectins have many interesting functions in terms of cell and molecular biology, the functions and mechanisms of galectins are complex and still unknown. Since the expression patterns of galectins differ across tissues, targeting galectins overexpressed in specific tissues may improve tissue selectivity. These features highlight the growing importance of galectin-targeted therapeutics, given their high selectivity and versatile roles across various applications. The medical applications of galectins are rapidly progressing from basic research to clinical applications, with further development expected in the future.

## Figures and Tables

**Figure 1 ijms-26-00790-f001:**
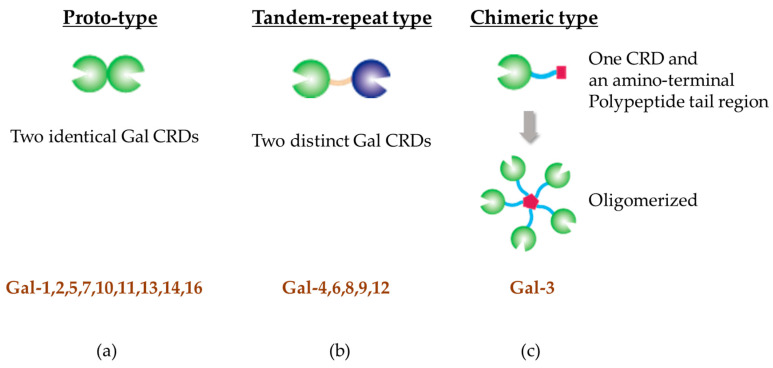
Galectins are classified into three subgroups, (**a**) proto-type, (**b**) tandem-repeat type, and (**c**) chimeric type, depending on the number and function of their CRDs.

**Figure 2 ijms-26-00790-f002:**
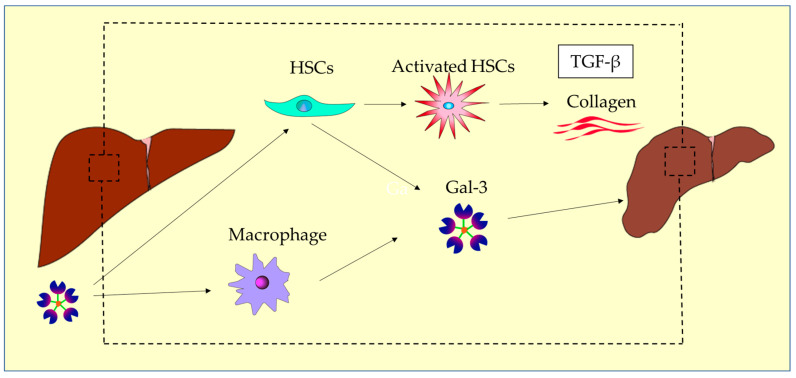
Summary of mechanisms of liver fibrosis.

**Figure 3 ijms-26-00790-f003:**
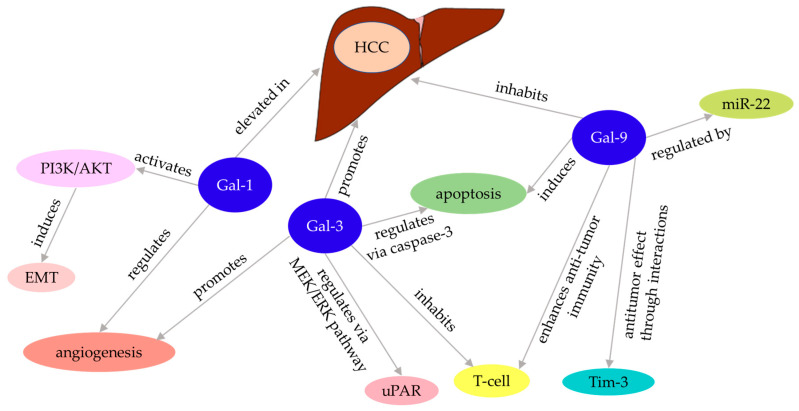
HCC and galectin interactions.

**Table 1 ijms-26-00790-t001:** Function of galectins in liver diseases.

Liver Disease	Related Galectins	Function and Clinical Significance	References
Chronic Hepatitis B	Gal-9	PD-L1-induced attenuation of antiviral cytokine release severely impairs the host’s immune response	[122]
		Independent prognostic markers of HBV-associated HCC	[123,124]
		The Tim-3/Gal-9 signaling pathway mediates T cell senescence	[125]
	Gal-3	Stimulates cytokine and chemokine production in macrophages via CD98.	[126]
		Supports HBV replication	[127,128,129]
		Contributes to chronic infection, and promotes fibrogenesis by reducing IL-10 production.	[127,128,129]
Chronic Hepatitis C	Gal-9	Expressed in hepatocytes and Kupffer cells	[130]
		Positively correlated with persistence of infection and progression of chronic liver disease	[131]
		Secretion is promoted by IFN-α and inhibits HCV infection	[132]
	Gal-3	Increased in HCV positive	[133]
Metabolic dysfunction-associated steatotic liver disease/steatohepatitis (MASLD/MASH)	Gal-3	Activates the TLR4/NLRP3 inflammasome, driving liver inflammation and lipid imbalance	[134]
		Elevated in obesity and diabetes, linked to disrupted glucose homeostasis.	[135]
		Promotes adipogenesis by aiding progenitor cell differentiation into adipocytes.	[135]
		Regulates inflammation and fibrosis, binds AGEs variably across tissues	[136]
	Ga-9	Regulates hepatic NKT cell homeostasis via Tim-3, inducing apoptosis and promoting proliferation,	[137]
		Improves diet-induced steatosis	[137]
Alcohol Associated Liver Disease	Gal-9	Increase Tim-3 levels	[138]
Autoimmune hepatitis (AIH)	Gal-9	Inhibits T helper 1 immune responses by binding to Tim-3 on CD4 effector cells	[139]
Primary biliary cholangitis (PBC)	Gal-3	Induces the production of inflammatory cytokines and damages biliary epithelial cells	[140]
Hepatocellular carcinoma	Gal-3	Inhibits tumor-reactive T cells and promotes tumor growth	[141]
		Promotes angiogenesis	[142]
		Inhibits tumor-reactive T cells	[143]
		inhibits uPAR expression and its downstream MEK/ERK signaling, reducing cell proliferation, migration, and invasion	[144]
	Gal-9	Induces apoptosis	[145]
		Boosts anti-tumor immunity by increasing Tim-3-expressing DCs and CD8+ T cells	[125]
		Induced apoptosis is inhibited by lactose, indicating the receptor requires β-galactoside glycosylation	[145]
		Expression is stimulated by interferon in HCC cells, and its antitumor effects are enhanced when its expression is restored in cells overexpressing miR-22	[146]
		Associated with a significantly shorter time to tumor recurrence.	[147]
	Gal-1	Interacts with mRNA that prefers binding to obstructive codons, thereby regulating angiogenesis	[148]
		Induction of epithelial mesenchymal transition	[149,150,151]

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
