# Peer review of "Galectins and Liver Diseases"

_ijms, 2025, doi:10.3390/ijms26020790_

Round 1
Reviewer 1 Report
Comments and Suggestions for Authors
The authors wrote a Review article about biologocal roles of galectin in liver diseases. This article is quite interesting for the field of liver diseases. Thus, I would like to recommned that this should be acceptable for the publication.
the review articles provide the advanced knowledges about a certain scientific field together with previously published papers. Based on this principle, this paper is quite interesting as the review article, and provides advanced knowledges to the readers.
Now, I have one question. It is well known that liver function has a close relationship with metabolic disorder. Now, I would like to ask you to describe how intrahepatic galectin is involved metabolic diseases.Author Response
Thank you very much for your positive and encouraging comments on our review article. We are delighted to hear that you find our work interesting and valuable for the field of liver diseases. Your recommendation for publication is greatly appreciated. We wish to express our appreciation to your insightful comments, which have helped us to improve the paper significantly. We would be happy to respond to any further questions or comments.
Comment: Now, I have one question. It is well known that liver function has a close relationship with metabolic disorder. Now, I would like to ask you to describe how intrahepatic galectin is involved metabolic diseases.
Response:
Thank you for your valuable question.
To the best of our knowledge, we could not find any studies directly linking intrahepatic galectins with metabolic diseases. Therefore, we focused on MASLD/MASH, as these liver diseases are strongly associated with metabolic disorders such as type 2 diabetes and obesity.
In particular, Gal-3 was discussed in detail, focusing on its association with obesity and glucose metabolism, which are related to the liver (page 9, line 315-330).
Gal-3 levels are higher in obese and diabetic individuals and have been shown to increase in association with uninhibited glucose homeostasis. Additionally, Gal-3 is thought to regulate adipogenesis by promoting the differentiation of progenitor cells into mature adipocytes.
Advanced glycation end products (AGEs) are a heterogeneous group of glycosylated proteins that accumulate with aging. Elevated AGE levels are observed in individuals with diabetes and liver cirrhosis. In diabetic patients, Gal-3 exhibits both AGE-binding-dependent and -independent functions. The AGE-dependent roles of Gal-3 differ across organs, reflecting tissue-specific variations in the AGE receptor system, with the liver playing a central role in the clearance and detoxification of AGEs from the bloodstream. Consequently, most studies regard Gal-3 as a marker of inflammation and fibrosis. However, research using animal models of metabolic disorders suggests that increased Gal-3 expression may serve as an adaptive response to tissue damage, aiding in the resolution of inflammation and preventing the progression to chronic inflammation.
We would like to express our sincere gratitude for your detailed and constructive review. Your suggestions have greatly enhanced the clarity and overall quality of our manuscript, and we are truly appreciative of the time and effort you dedicated to reviewing our work. Thank you once again for your invaluable comments.
Reviewer 2 Report
Comments and Suggestions for Authors
the authors reviewed the complex and multifaceted realm of gelatins in a very systematic and concise fashion. I have some minor points to note. To the best of my knowledge there are total 16 numbers of gelatins discovered till date. I think the authors took the number of total 19 galectins from the original reference where they somehow mistyped the number. If the author believes that there are total 19 galectins indeed, I'll strongly recommend to categorize them (16-19) into types (proto, tandem or chimeric types). Lines 73 to 79 seems very repetitive specially when the authors talks about tissue expression of galectin-1, 3 and 12.
I also encourage to include the following references. Dumic, J., Dabelic, S.; Flogel, M. Galectin-3: An open-ended story, Biochemica et Biophysica Acta, 2006, 1760, 616-636.; S. Sciacchitano, L.; Lavra, A.; Morgante, A.; Ulivieri, A.; Magi, F.; De Francesco, G. P.; Morgante, Bellotti, C.; Salehi, L. B.; Ricci, A. Galectin-3: One Molecule for an Alphabet of Diseases, from A to Z. Int. J. Mol. Sci., 2018, 19.; Hara, A.; Niwa, M.; Noguchi, K.; Kanayama, T.; Niwa, A.; Matsuo, M.; Hatano, Y.; Okada, H.; Tomita, H. Galectin-3 as a Next-Generation Biomarker for Detecting Early Stage of Various Diseases. Biomolecules, 2020, 10.
Author Response
Thank you very much for your valuable comments and suggestions. We have carefully addressed all your feedback, which has been instrumental in significantly improving the quality of our manuscript. We greatly appreciate your insightful remarks, which provided a fresh perspective and helped refine our work. In particular, the references you recommended were extremely helpful in enhancing the depth and clarity of our review. If you have any further questions or additional comments, we would be more than happy to address them. Thank you again for your thoughtful comments.
Comment: To the best of my knowledge there are total 16 numbers of gelatins discovered till date. I think the authors took the number of total 19 galectins from the original reference where they somehow mistyped the number. If the author believes that there are total 19 galectins indeed, I'll strongly recommend to categorize them (16-19) into types (proto, tandem or chimeric types).
Response:
Thank you for your valuable comment.
In this study, we have decided to follow the numbering system for galectins as described in the work by Ralf Jacob et al (Jacob, R.; Gorek, L.S. Intracellular Galectin Interactions in Health and Disease. Semin Immunopathol 2024, 46, 4.). In their publication, they adopted the classification of the HUGO Gene Nomenclature Committee (HGNC), wherein the galectins identified in the human genome are classified as Galectin-1, -2, -3, -4, -7, -7B, -8, -9, -9B, -9C, -10, -12, -13, -14, and -16 (gene group: Galectins (LGALS))(page 2, line 54-56).
Comment: Lines 73 to 79 seems very repetitive specially when the authors talks about tissue expression of galectin-1, 3 and 12.
Response:
Thank you for your comment. We have revised lines 73 to 79 to reduce repetition, particularly in the discussion of tissue expression of galectin-1, -3, and -12. We appreciate your feedback.
Comment:I also encourage to include the following references.
Response:
Thank you for your suggestion. We have reviewed the references you proposed and included them in our review where relevant. We appreciate your recommendation, as these references provide valuable insights into the topic.
The references you suggested were extremely helpful in understanding galectins. We have cited the key points from each of these references. Gal-3 is a unique molecule that can form pentamers and has multifunctional biological properties, playing a crucial role in various physiological and pathological processes. We have added the reference you recommended to our review (page 2, line 73-74).
Dumic, J., Dabelic, S.; Flogel, M. Galectin-3: An open-ended story, Biochemica et Biophysica Acta, 2006, 1760, 616-636.
We have added the reference you recommended to our review to highlight the essential role of Gal-3 (page 2, line 74-76).
- Sciacchitano, L.; Lavra, A.; Morgante, A.; Ulivieri, A.; Magi, F.; De Francesco, G. P.; Morgante, Bellotti, C.; Salehi, L. B.; Ricci, A. Galectin-3: One Molecule for an Alphabet of Diseases, from A to Z. Int. J. Mol. Sci., 2018, 19.
We have already referenced this article (page 2, line 59-61). Although we have only cited a small portion this time, reviewing it again was helpful in deepening our understanding of galectins.
Hara, A.; Niwa, M.; Noguchi, K.; Kanayama, T.; Niwa, A.; Matsuo, M.; Hatano, Y.; Okada, H.; Tomita, H. Galectin-3 as a Next-Generation Biomarker for Detecting Early Stage of Various Diseases. Biomolecules, 2020, 10.
We would like to express our sincere gratitude for your detailed and constructive review. Your suggestions have greatly enhanced the clarity and overall quality of our manuscript, and we are truly appreciative of the time and effort you dedicated to reviewing our work. Thank you once again for your invaluable comments.
Reviewer 3 Report
Comments and Suggestions for Authors
Dear Authors,
Your manuscript presents a comprehensive review of an important topic. It is also written in detail and the schemes are adequate. However, there are previously published works on the same topic (one of them is cited as reference 171), and nowhere is your new scientific contribution to this topic highlighted. From this reason, I believe that major changes are necessary before further consideration of your work for publication in the International Journal of Molecular Sciences.
Some of published paper with the same topic:
Mackinnon AC, Tonev D, Jacoby B, Pinzani M, Slack RJ. Galectin-3: therapeutic targeting in liver disease. Expert Opinion on Therapeutic Targets. 2023 Sep 2;27(9):779-91.
An Y, Xu S, Liu Y, Xu X, Philips CA, Chen J, Méndez-Sánchez N, Guo X, Qi X. Role of Galectins in the Liver diseases: a systematic review and Meta-analysis. Frontiers in Medicine. 2021 Oct 27;8:744518.
Ezhilarasan D. Unraveling the pathophysiologic role of galectin‐3 in chronically injured liver. Journal of Cellular Physiology. 2023 Apr;238(4):673-86.
Kram M. Galectin-3 inhibition as a potential therapeutic target in non-alcoholic steatohepatitis liver fibrosis. World Journal of Hepatology. 2023 Feb 2;15(2):201.
Authors in the manuscript line 45 an 46 stated: "However, little is known about the involvement of galectins in liver disease. Here, we review the functions of various galectins and evaluate their roles in liver diseases."Firstly, If little is known about something, it is impossible to expand that knowledge by review paper. Knowledge can be systematized and that contribution can be great. However, there are review papers examining the connection between galactins and liver disease, and it is necessary that they be mentioned and cited in the adequate manner. This approach is ethically and scientifically justified. Also, it is necessary for the authors to emphasize their originally contribution in the literature review. Is it more detailed, does it include some new aspects, etc. In my opinion, it is necessary to add a paragraph after lines 45 and 46 in which the described aspect will be stated.
Author Response
Thank you very much for your valuable comments and suggestions. We have carefully addressed all your feedback, which has been instrumental in significantly improving the quality of our manuscript. We greatly appreciate your insightful remarks, which provided a fresh perspective and helped refine our work. In particular, the references you recommended were extremely helpful in enhancing the depth and clarity of our review. If you have any further questions or additional comments, we would be more than happy to address them. Thank you again for your thoughtful comments.
Indeed, as you pointed out, our review may seem similar to previously published papers. However, we believe that our current review provides the most comprehensive explanation of liver diseases compared to any previous reviews. For instance, in reference 171, the roles of Galectins in individual liver diseases, except for hepatocellular carcinoma, are only briefly discussed. Furthermore, emerging topics such as liver diseases, Galectins, and the Gut-Liver Axis are not addressed. While our review covers liver diseases in a general sense, it uniquely offers a relatively detailed and comprehensive discussion of the relationship between Galectins and the entire spectrum of liver diseases, making it a valuable contribution and well-suited as a review article.
We completely agree with your opinion, and we have incorporated the suggested content into a paragraph added after lines 45 and 46 (page 2, line 45-50, introduction). As mentioned, this addition provides a comprehensive explanation of the described aspect and aligns well with the overall structure of the review. Thank you for your valuable comments.
We have added a work published this year that discusses new findings regarding Gal-9 and the recurrence of hepatocellular carcinoma (page 11, line 397-405, 8.7.Hepatocellular Carcinoma and Galectins).
Additionally, we have referenced new reference on the Gut-Liver Axis and provided a clear explanation (page 7, line256-259, 7. Galectins and the Gut-Liver Axis).
The references you suggested were extremely helpful in understanding galectins. We have cited the key points from each of these references. We have included a reference to the work you introduced. The work mentions Gal-3 as a marker for liver fibrosis, and we have discussed its content in our review (reference181, page 12, line 424-426, 9.1. Liver fibrosis screening). Mackinnon AC, Tonev D, Jacoby B, Pinzani M, Slack RJ. Galectin-3: therapeutic targeting in liver disease. Expert Opinion on Therapeutic Targets. 2023 Sep 2;27(9):779-91.
We have learned that Gal-3 amplifies the production of extracellular matrix and fibrogenesis through autocrine signaling pathways, and we have added this information to our review (page 6, line 245-246, 6. Liver Fibrosis). Ezhilarasan D. Unraveling the pathophysiologic role of galectin‐3 in chronically injured liver. Journal of Cellular Physiology. 2023 Apr;238(4):673-86.
We have already referenced these articles. While we have only cited a small portion this time, reviewing them again has been helpful in deepening our understanding of galectins;
An Y, Xu S, Liu Y, Xu X, Philips CA, Chen J, Méndez-Sánchez N, Guo X, Qi X. Role of Galectins in the Liver diseases: a systematic review and Meta-analysis. Frontiers in Medicine. 2021 Oct 27;8:744518.
Ezhilarasan D. Unraveling the pathophysiologic role of galectin‐3 in chronically injured liver. Journal of Cellular Physiology. 2023 Apr;238(4):673-86.
We would like to express our sincere gratitude for your detailed and constructive review. Your suggestions have greatly enhanced the clarity and overall quality of our manuscript, and we are truly appreciative of the time and effort you dedicated to reviewing our work. Thank you once again for your invaluable comments.
Round 2
Reviewer 3 Report
Comments and Suggestions for Authors
Dear Authors,
You have improved your manuscript.
Good luck!